# Trends in cervical cancer and its precursor forms to evaluate screening policies in a mid-sized Northeastern Brazilian city

**Marcela Sampaio Lima**[1,2☉], **Érika de Abreu Costa Brito**[1,2☉], **Hianga Fayssa Fernandes Siqueira**[1,2☉], **Marceli de Oliveira Santos**[3☉], **Angela Maria da Silva**[1,2☉], **Marco Antonio Prado Nunes**[1,2☉], **Hugo Leite de Farias Brito**[1,2☉], **Marcia Maria Macedo Lima**[2☉], **Rosana Cipolotti**[1,2☉], **Carlos Anselmo Lima**[1,2,4☉]*

**1** Graduate Program in Health Sciences, Federal University of Sergipe, Aracaju, Sergipe, Brazil, **2** University Hospital, EBSERH, Federal University of Sergipe, Aracaju, Sergipe, Brazil, **3** CONPREV, Brazilian National Cancer Institute, Rio de Janeiro, Rio de Janeiro, Brazil, **4** Aracaju Cancer Registry, Aracaju, Sergipe, Brazil

☉ These authors contributed equally to this work.
* carlos.a.lima@ufs.br

**Data Availability Statement:** All relevant data are within the manuscript and its Supporting Information files.

## Abstract

Cervical cancer is a health issue that disproportionately affects developing countries, where the Papanicolaou test (Pap smear) remains an important screening tool. Brazilian government recommendations have focused screening on the female population aged from 25 to 64 years old. In this study, we examined the incidence and mortality rates of invasive cervical cancer lesions and the incidence rates of in situ precancerous cervical lesions, aiming to calculate their respective statistics over time in a mid-sized Brazilian city, Aracaju. The 1996–2015 database from the Aracaju Cancer Registry and Mortality Information System was used to calculate age standardized rates for all invasive cervical tumors (International code of diseases, ICD-10: C53) and preinvasive cervical lesions (ICD-10: D06) in the following patient age ranges; ≤ 24, 25–34, 35–44, 45–54, 55–64 and ≥ 65 years old. We identified 1,030 cancer cases, 1,871 in situ lesions and 334 deaths. Using the Joinpoint Regression Program, we calculated the annual percentage incidence changes and our analyses show that cervical cancer incidence decreased up to 2008, increased up to 2012 and decreased again thereafter, a significant trend in all age groups from 25 years. The incidence of precursor lesions increased from 1996 to 2005 and has since decreased, a result significant in all age groups until 64 years. Cervical cancer mortality has decreased by 3.8% annually and trend analysis indicates that Pap smears have been effective in decreasing cancer incidence and mortality. However, recent trends shown here show a decreasing incidence of in situ lesions and may indicate either a real decrease or incomplete catchment. Thus, we suggest health policies should be re-considered and include sufficient screening and HPV vaccination strategies to avoid cervical cancer resurgence in the population.

**Funding:** CAL This research was conducted with the partial support of a Research Development Grant from the Fundação de Apoio à Pesquisa e à Inovação Tecnológica do Estado de Sergipe - FAPITEC/SE Protocolo: 019.203.00961/2018-2.

**Competing interests:** The authors have declared that no competing interests exist.

## Introduction

Cervical cancer still has high rates of incidence and mortality, despite the epidemiological transition having occurred in many countries. Statistics show regional variation and are dependent on the human development index of the area[1]. The main causative agent of cervical cancer is Human Papilloma Virus (HPV) types 16 and 18. As such, it has been inferred that HPV vaccination, together with screening using the Papanicolaou test (Pap smear) and HPV DNA identification[2][3][4], is important to reduce incidence and mortality rates. However, it is necessary to evaluate the outcomes from past and current health programs for future comparison with programs that include HPV vaccination.

The two major histological subtypes of cervical cancer, squamous cell carcinoma and adenocarcinoma, are equally dependent on HPV infection and the diagnosis and treatment of their preinvasive forms is crucial to blocking transformation into invasive carcinoma[5].

Screening for cervical cancer has the greatest impact in the 25–65-year-old age group[6]. In Brazil, government protocols in 1998 recommended to start screening patients when they are between 35 to 49 years old. From 2011, this was updated to begin screening people from 25 years old and stop when they are 64, after at least two negative tests in the five years prior, and the HPV vaccination program began in 2014[7]. However, varying regional development has led to differences in patient recruitment for screening, access to health services and the quality of mortality data. These issues hinder evaluations to assess the impact of screening on the incidence of cervical cancer and associated mortality in Brazil[8].

The purpose of this study is to estimate the impact of Pap smear screening on the trends in incidence of invasive and preinvasive cervical cancer lesions, and associated mortality, to determine whether public policies have been effective. We hypothesize that increased identification of preinvasive lesions and subsequent intervention will lead to reduced incidence of invasive neoplasms, and consequently, reduced mortality rates. The assessment of vaccination results, which might also improve control, needs to be undertaken.

## Materials and methods

The population used in this study was that of the municipality of Aracaju, Sergipe, Brazil estimated at 648,939 in 2018, with approximately 320,000 women at risk per year.

We used data from the Aracaju Cancer Registry from 1996 to 2015 to calculate incidence rates. The cancer registry actively collects information from Aracaju hospital records, accesses the databases of all pathology and cytology laboratories in Aracaju, and links to several official health information databases, including information on cervical cancer screening. To determine mortality rates, we used all-cause mortality data from the Brazil Mortality Information System for the same period. The Aracaju Cancer Registry was established in 1998 and contains a comprehensive and internationally validated dataset from this date.

The registry records in situ lesions as invasive if there have been two positive diagnoses in less than one year. We did not calculate mortality for carcinomas in situ because the risk of death is only substantial at high ages, independent of age at diagnosis, indicating alternative primary causes. We have used all-cause mortality, not stratifying cause-specific deaths.

We included all cases of cervical cancer and preinvasive lesions according to the International Classification of Diseases, 10th Revision (ICD-10), C53, D06 and N87.2. Invasive lesions were defined as squamous cell carcinoma (morphology: 8010–8560, 8050–8052, 8070–8076, 8082 and 8123); adenocarcinoma (morphology: 8140, 8231, 8255, 8260, 8310, 8380, 8384, 8430, 8440, 8441, 8450, 8460, 8480, 8481, 8490, 8570, 9110); and others (morphology: 8000, 8010, 8013, 8015, 8020, 8041, 8090, 8200, 8246, 8560, 8574, 8720, 8800, 8810, 8830, 8890, 8900, 8910, 8920, 8933, 8935, 8950, 8951, 8980, 9473). In situ lesions were defined for morphologies

8010–8560). Malignant neoplasms of the uterine body (ICD-10 C54) and malignant neoplasms of an unspecified portion of the uterus (ICD-10 C55), were excluded from the analysis.

We calculated the age standardized rates (ASRs) of the whole population for incidence and mortality in each year using the direct method of standardization, which combines the population's age specific rates with the composition of a standard population; here, the number of individuals by five-year age groups to a total of 100,000. This is a correction to the age structure of the world population. It resulted in weighted age-specific rates to reflect the number of events if the population analyzed had the same age distribution[9] Population censuses and estimates have been obtained from the Brazilian Institute of Geography and Statistics (IBGE). To calculate the age-specific rates, we defined for this study as follows: ≤ 24, 25–34, 35–44, 45–54, 55–64, ≥ 65 years. Therefore, we were able to separately analyze the Brazilian Ministry of Health's priority patient screening group of 25–64 years old.

To measure incidence and mortality changes, we have calculated the annual percentage change (APC), the average annual percentage change (AAPC) and their corresponding 95% confidence intervals (95% CI) using the Joinpoint Regression Program[10]. The model selection was performed by the Monte Carlo Permutation Test, which determines the p-value from the permutation distributions derived from F-statistic as a goodness-of-fit measure[11].

This research project was approved by the Research Ethics Committee of the Federal University of Sergipe and all methods were executed in accordance with the relevant guidelines and regulations. We have used patient anonymized databases and consequently obtaining informed consent was infeasible. For this, we have been given an exemption by the ethics committee, as specified in Resolution number 466, December 12th, 2012, of the Ministry of Health of Brazil.

## Results

The Aracaju Cancer Registry recorded 1,060 cases of cervical cancer, 1,997 of carcinoma in situ and 354 associated deaths, from 1996 to 2015 (obtained from the Brazil Mortality Information System). Among the invasive neoplasms, 80% were squamous cell carcinomas, as shown in Fig 1.

Table 1 shows the annual number of cases, the age standardized rates of incidence and their respective confidence intervals. Overall, data are stable year to year.

When analyzing all ages combined, the ASR curve for invasive tumor incidence (Fig 2) presented a decreasing trend for incidence from 1996 to 2015. However, it fitted three joinpoints with non-significant trends: a decreasing trend from 1996 to 2008, a rising trend from 2008 to 2012, and a decreasing trend from 2012 on. The AAPC for incidence of invasive neoplasms over the whole period was −6.2 (95% CI: −7.9; −4.5). All individual age groups demonstrated decreasing incidence trends over time. However, mortality trends were statistically significant only in the 45–54-year-old age group. We separately analyzed the incidence of invasive lesions for the middle-aged adult group (45–64 years old), which included the most cases. Our analyses did not fit any joinpoints in the model and the APC was determined as −7.1 (95% CI: −9.2; −5.1).

For incidence of carcinomas in situ, we observed an upward trend in the ASR for all ages with AAPC of 13.3 (5.8; 21.3) until 2005, followed by a downward curve (Table 2, Fig 2), with AAPC of −4.8 (−8.5; −0.9). All age groups up to 64-year-olds presented similar curves to ASR for all ages, with one joinpoint, whereas the data for ≥65-year-olds were fitted with no joinpoints.

Cervical cancer mortality decreased over time (Table 2, Fig 2) across all ages with an AAPC −3.8 (95% CI: −5.9; −1.7). Only the 45–54 age group showed a statistically significant decrease,

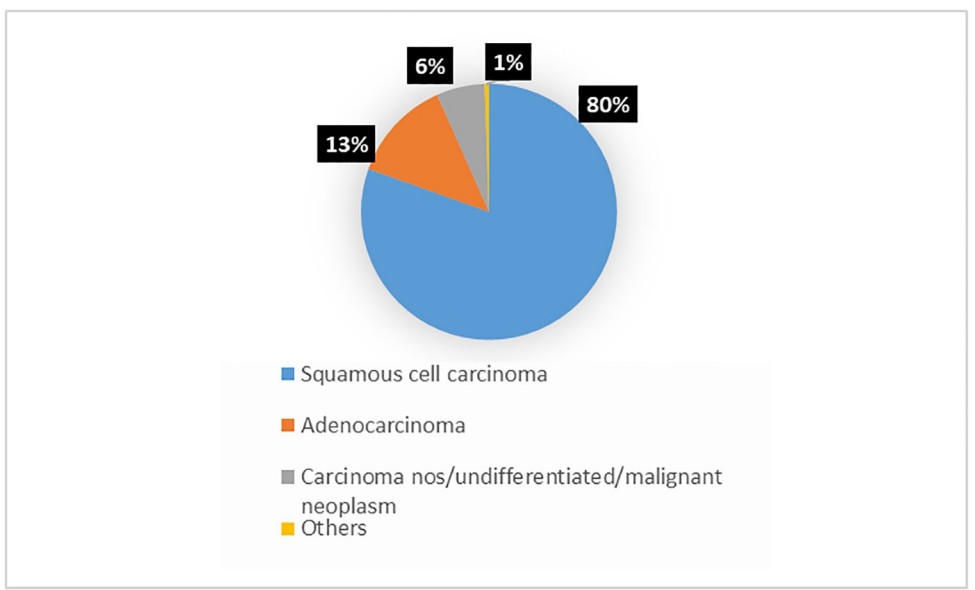

**Fig 1. Numbers and percentage of cases of cervical cancer by morphology.**

**Table 1. Annual age-standardized rates with confidence intervals; 1996–2015.**

| | Incidence, invasive | | | Mortality | | | Incidence, in situ | | |
|---|---|---|---|---|---|---|---|---|---|
| Year | N(1,060) | ASR | 95%CI | N(354) | ASR | 95%CI | N(1,997) | ASR | 95%CI |
| 1996 | 59 | 32.3 | 24.1; 40.6 | 15 | 8.1 | 4.0; 12.3 | 40 | 1.19 | 12.1; 23.0 |
| 1997 | 66 | 36.1 | 27.4; 44.9 | 12 | 6.1 | 2.6; 9.5 | 44 | 17.6 | 12.4; 22.8 |
| 1998 | 53 | 28.2 | 20.6; 35.8 | 16 | 8.8 | 4.5; 13.1 | 58 | 23.0 | 17.1; 28.9 |
| 1999 | 59 | 30.0 | 22.4; 37.7 | 20 | 10.1 | 5.7; 14.5 | 50 | 20.1 | 14.5; 25.7 |
| 2000 | 55 | 24.1 | 17.7; 30.4 | 24 | 11.4 | 6.8; 16.0 | 47 | 17.4 | 12.4; 22.3 |
| 2001 | 75 | 34.5 | 26.0; 42.3 | 14 | 6.2 | 2.9; 9.4 | 75 | 27.7 | 21.4; 33.9 |
| 2002 | 75 | 33.0 | 25.5; 40.4 | 16 | 6.9 | 3.5; 10.2 | 91 | 33.0 | 26.2; 39.8 |
| 2003 | 64 | 29.5 | 22.2; 36.7 | 22 | 10.3 | 6.0; 14.6 | 49 | 17.3 | 12.4; 22.1 |
| 2004 | 54 | 24.3 | 17.8; 30.7 | 19 | 8.4 | 4.6; 12.1 | 123 | 43.3 | 35.6; 50.9 |
| 2005 | 57 | 23.6 | 17.5; 29.8 | 21 | 9.8 | 5.6; 14.1 | 152 | 51.5 | 43.3; 59.7 |
| 2006 | 58 | 24.5 | 18.2; 30.8 | 14 | 5.8 | 2.8; 8.9 | 163 | 55.1 | 46.7; 63.6 |
| 2007 | 37 | 12.9 | 8.7; 17.0 | 14 | 5.1 | 2.5; 7.8 | 121 | 37.4 | 30.7; 44.0 |
| 2008 | 30 | 9.6 | 6.2; 13.1 | 21 | 7.1 | 4.1; 10.1 | 126 | 36.8 | 30.4; 43.3 |
| 2009 | 33 | 10.3 | 6.8; 13.8 | 16 | 5.4 | 2.7; 8.0 | 95 | 27.6 | 22.0; 33.1 |
| 2010 | 47 | 14.5 | 10.4; 18.7 | 22 | 6.7 | 3.9; 9.5 | 153 | 40.2 | 33.8; 46.5 |
| 2011 | 50 | 15.1 | 10.9; 19.3 | 13 | 3.7 | 1.7; 5.7 | 116 | 30.7 | 25.1; 36.3 |
| 2012 | 52 | 15.8 | 11.5; 20.1 | 21 | 6.7 | 3.9; 9.6 | 114 | 29.6 | 24.1; 35.0 |
| 2013 | 47 | 12.8 | 9.1; 16.5 | 16 | 4.1 | 2.1; 6.2 | 120 | 29.1 | 23.9; 34.4 |
| 2014 | 54 | 14.1 | 10.3; 17.8 | 18 | 4.8 | 2.6; 7.1 | 135 | 32.7 | 27.1; 38.2 |
| 2015 | 35 | 9.1 | 6.1; 12.2 | 20 | 5.2 | 2.9; 7.3 | 125 | 30.2 | 24.9; 35.5 |

N: number of cases; ASR: age-standardized rate; 95%CI: 95% confidence interval.

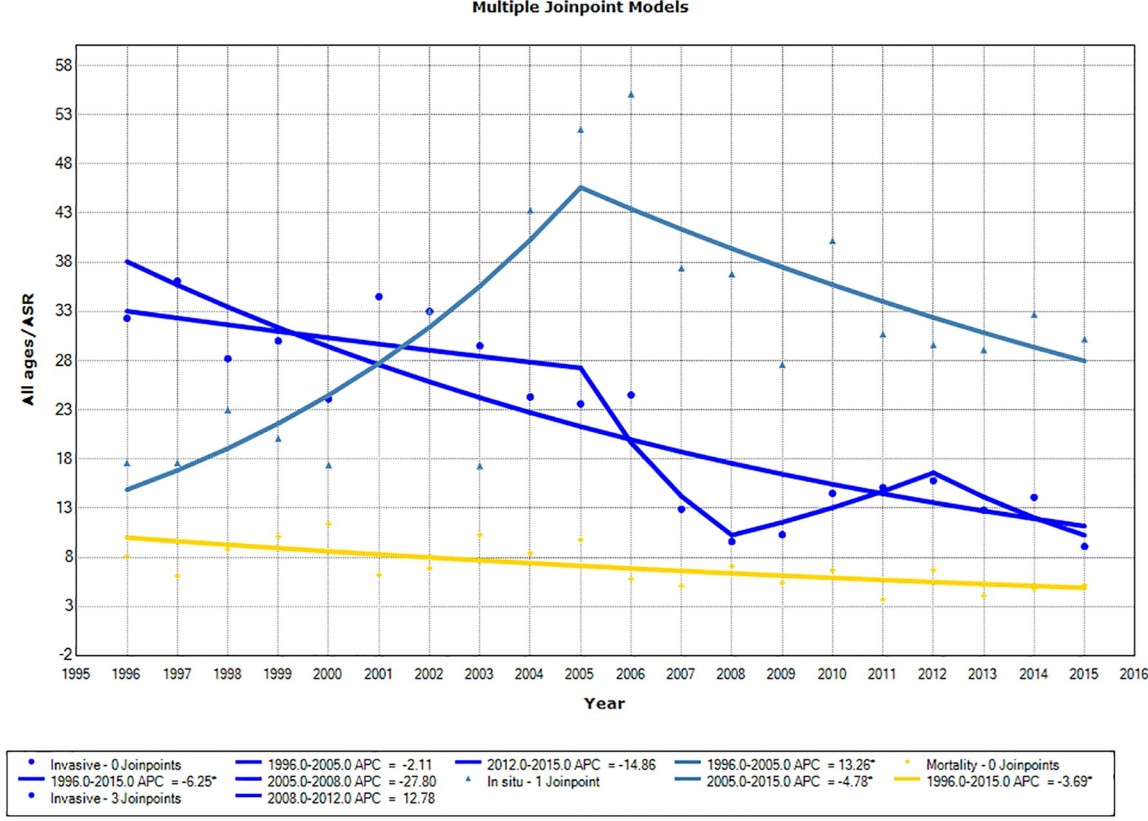

**Fig 2. Trends in age-standardized incidence and mortality rates for cervical cancer.** The age-standardized rate curves are shown (ASR) for all ages, with 3 joinpoints, demonstrating Annual Percent Change (APC) with no joinpoint, expressing Average Annual Percent Change (AAPC) (dark blue); carcinoma in situ incidence curve (lighter blue); and mortality curve (yellow).

with an APC of −4.2 (95% CI: −7.6; −0.6). Data from the other age groups show decreases in mortality over time, but these trends are statistically non-significant.

In this study, we found that the incidence rates (measured as ASRs) of invasive cervical cancer decreased until the year 2008, have shown a non-significant upward trend up to 2012 and thereafter a decreasing trend again. In some age groups this decrease in incidence has been maintained throughout the time series. Conversely, precursor lesions of cervical cancer showed increased incidence rates up to 2005, followed by a decrease. Fig 3 shows a comparison of the incidence of carcinoma in situ and invasive carcinoma, showing an average difference of 15 years between the peak ages of incidence.

## Discussion

Our findings suggest that the health policies implemented in Brazil to combat cervical cancer may be beneficial in reducing its instance and associated mortality. Our analysis shows an increasing incidence of precursor lesions up to 2005, which suggests improved early detection. However, public health managers should continue to optimize screening measures. Without this the detection of precursor lesions can decrease, as was observed in our study population, which could potentially lead to increased occurrence of invasive lesions. This might negatively affect cervical cancer mortality in the future. We cannot be certain whether our observed decrease in precursor lesion incidence was caused by incomplete catchment or true

**Table 2. Output of joinpoint analyses of carcinoma in situ incidence, invasive carcinoma incidence and mortality from cervical carcinoma data.** Joinpoints and APCs of ASRs with associated 95% CIs are shown, separated into age-specific groups.

| Age group | Incidence in situ | | Incidence inv | | Mortality | |
|---|---|---|---|---|---|---|
| | JP seg | APC (95% CI) | JP seg | APC (95% CI) | JP seg | APC (95% CI) |
| All | 1996–2005 | 13.3* (5.8; 21.3) | 1996–2005 | -2.1 (-6.2;2.1) | 1996–2015 | -3.8* (-5.9; -1.7) |
| | 2005–2015 | -4.8* (-8.5; -0.9) | 2005–2008 | -27.8 (-57.8; 23.4) | | |
| | | | 2008–2012 | 12.8 (-15.1; 49.8) | | |
| | | | 2012–2015 | -14.9 (-35.9; 13.1) | | |
| ≤ 24 | 1996–2006 | 24.3* (12.8; 37.0) | NF | NF | NF | NF |
| | 2006–2015 | -6.8 (-13.2; 0.2) | | | | |
| 25–34 | 1996–2005 | 14.5* (4.9; 25.0) | 1996–2015 | -6.8* (-9.5; -4.0) | 1996–2015 | -4.0 (-8.0; 0.1) |
| | 2005–2015 | -2.0 (-6.6; 2.9) | | | | |
| 35–44 | 1996–2005 | 10.8* (3.5; 18.8) | 1996–2015 | -4.4* (-7.0; -1.6) | 1996–2015 | -2.1 (-6.1; 2.2) |
| | 2005–2015 | -6.3* (-10.5; -1.9) | | | | |
| 45–54 | 1996–2006 | 9.7 (-0.5; 21.1) | 1996–2015 | -6.0* (-8.2; -3.7) | 1996–2015 | -4.2* (-7.6; 0.6) |
| | 2005–2015 | -7.5* (-12.9; -1.8) | | | | |
| 55–64 | 1996–2015 | -2.2 (-6.2; 2.0) | 1996–2015 | -7.6* (-10.6; -4.5) | 1996–2015 | -3.4 (-7.8; 1.3) |
| ≥ 65 | 1996–2015 | 2.5 (-2.3; 7.5) | 1996–2015 | -5.4* (-8.5; -2.1) | 1996–2015 | -3.0 (-6.6; 0.8) |

APC: annual percent change; ASR: age-standardized rate; CI: confidence interval; JP seg: time range segment; Incidence in situ: carcinoma in situ incidence; Incidence inv: invasive carcinoma incidence; NF: model not fitted.

*Significant APC; p ≤ 0.05.

diminished incidence. Our observation of a reduction in incidence of cervical carcinoma in situ from 2005 and a rise in the incidence of invasive cervical neoplasms from 2008 suggests that changes in the screening policies, compliance, or in the cancer notification system have occurred. However, decreasing observed incidence data from 2012 on could be seen as

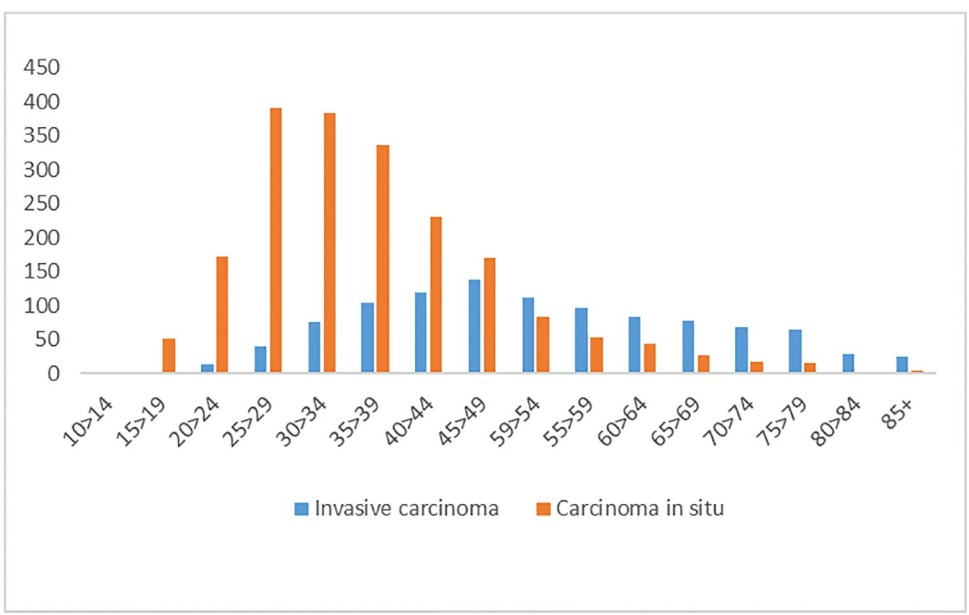

**Fig 3. Number of cases of carcinoma in situ and invasive carcinoma of the cervix, across age groups.**

evidence for decreased true incidence. To our knowledge, no solid evidence for such a decrease has been reported.

The importance of detecting precursor lesions is stressed by Moreno et al.[12] who assessed that 36% of cases of carcinoma in situ would progress to invasive if they remained without treatment. Monitoring outcomes after diagnosis of carcinoma in situ is unethical, given the risk of progression. As such, the rate of progression from in situ to invasive is always an estimation, with one study showing progression rates between 31% and 50% within 30 years[13]. In this study, the progression rate after appropriate treatment was just 0.7%, which is why treating carcinoma in situ is paramount.

In Brazil, the National Cancer Institute (INCA) has developed the National Cervical Cancer Control Program (PNCCC), which was implemented in 2001. This program aims to guarantee access to cervical examination, diagnosis and preventative treatment of precursor lesions for women of the prioritized age group (25–64 years old). In 2014, the Ministry of Health included the HPV vaccine in the National Vaccination Program. The vaccine is highly effective against HPV types 6, 11, 16 and 18[14]. HPV types 16 and 18 account for about 70% of cases of cervical cancer worldwide[15]. At the outset, the target population for vaccination was composed of girls aged 11–13 years. In subsequent years, the target age group has been expanded and boys included in the vaccine program[16]. It is hoped that the improvements in the screening strategy, together with HPV prophylaxis, will reduce cervical cancer mortality rates in the Brazilian population. Comparative trend studies will be needed to assess this.

Estimates of cervical cancer incidence for 2020 in Brazil[17] give ASRs of 12.6 and 10.1 / 100,000 in states and capitals, respectively. In the northeastern region of Brazil, which is less developed, ASRs are estimated at 16.1 and 10.1 / 100,000 in states and capitals, respectively. In this study, the incidence ASR for 2015 is lower than those estimated for Brazil. This could indicate a continued rising trend of incidence, which may be maintained if early detection strategies are not re-evaluated. Some authors mention that a decrease in cervical cancer incidence could occur mainly as a consequence of a decrease in the incidence of squamous cell carcinoma[18][19], which reflects improved control of HPV infection, which could be further improved with vaccination[20][21].

Screening policies are designed to decrease mortality[22][23] and a decrease in mortality throughout the time series in this study suggests that this has been successful. Sousa et al.[10] evaluated the mortality trend for cervical cancer in the state of Rio Grande do Norte, also located in the northeast of Brazil. They also observed a decreasing trend in ASR of 5.95 deaths/ 100,000 women per year in the period 2006–2010, and they predicted 3.67 deaths/ 100,000 women per year for the period 2026–2030. The ASR trend curves for invasive carcinoma incidence, which show an increasing trend in the last six years, and those of carcinoma in situ incidence, which show a decreasing trend in the last nine years of the series, can be compared with the mortality curves across the same period. From this, it seems possible that these shifting incidence patterns might lead to increased mortality in the future. Our study therefore should alert public managers that, if improvements in the screening process are not implemented, cervical cancer mortality could approach past levels.

The role of screening in decreasing cervical cancer mortality is well defined by Landy et al. [24], who predicted that mortality would be four times higher in women aged 35–49 years without screening, and also higher in the 50–64 age group. The opposite is predicted if screening were comprehensive; with mortality predicted to halve in women aged 35–49 years and decrease by even more in those aged 50–64 years. However, in Brazil, there has been a decrease in mortality in more developed regions and an increase in less developed ones[25]. This suggests differences in access to the means of prevention and treatment of precursor and invasive forms of cervical neoplasms.

Several studies have reported results similar to ours, indicating that cervical cancer screening has declined in recent years[26][27][28][29]. Control policies need to be re-evaluated to include new strategies such as the systematic incorporation of HPV vaccination[30], especially in less developed regions of the country. In the findings of Sreedevi et al. in India, the peak age of incidence is from 55 to 59 years[31], which indicates that this age group should be specifically targeted to improve screening strategies and participation. However, all age groups in this study presented decreasing trends in incidence of cervical cancers.

Oke et al.[32] found that cervical cancer incidence in the UK population has increased by more than 150% since 1980. Conversely, mortality has decreased by 69% in the same period, and continues to decline. Interestingly, they verified that diagnosis of cervical carcinoma in situ accounted for all this incidence increase, while rates of invasive cervical neoplasms have almost halved since 1980. Oke et al. suggested that efficient and comprehensive screening programs are more likely to detect indolent disease in the asymptomatic population. The population in our study has greater difficulty in accessing prevention programs and treatment in the public health network and we have found a different scenario. In Brazil, the incidence of in situ cervical lesions has been decreasing since 2006, while that of invasive neoplasms increased from 2006 to 2012 and decreased thereafter, although mortality has still decreased.

We have identified two limitations in this study. First, because of regional differences in screening strategies and in access to health services the results of the present study cannot be completely extrapolated to the other areas Brazil. Second, mortality rates are influenced by the quality of death certificate completion and the presence of a high percentage of undetermined causes of death in this dataset is likely to have affected mortality rates and trends. Recently, however, undetermined causes of death have dropped to acceptable levels.

## Conclusions

In summary, we have found that incidence rates and mortality for cervical cancer have decreased in the years since screening strategies have been in operation, in the population of study. the recent trends showing a decreasing incidence of cervical precursor lesions may indicate either that their incidence has diminished or that the detection incidence has diminished without a change in underlying lesion occurrence. If the latter is true, invasive tumor incidence rates could return to higher levels, which is likely to impact the future mortality associated with cervical cancer. In this case, Brazilian health policies should be re-considered and include strategies to screen every woman in a defined age group and to assess and improve the rates of HPV vaccination.

## Supporting information

**S1 Data.**
(XLSX)

## Acknowledgments

We thank the personnel of the Cancer Registry for their work in collecting data and preparing the database for this research. They are; José Erinaldo Lobo de Oliveira, Elma Santana de Oliveira, Maria das Graças Prata França, Sueli Pina Vieira, Marina Kobilsek, Analeide Rezende e Cecília Ferreira.

We thank Edanz Group (https://en-author-services.edanzgroup.com/) for editing a draft of this manuscript.

## Author Contributions

**Conceptualization:** Marcela Sampaio Lima, Marceli de Oliveira Santos, Carlos Anselmo Lima.

**Formal analysis:** Marcela Sampaio Lima, Carlos Anselmo Lima.

**Methodology:** Marceli de Oliveira Santos, Carlos Anselmo Lima.

**Validation:** Érika de Abreu Costa Brito, Hianga Fayssa Fernandes Siqueira, Angela Maria da Silva, Marco Antonio Prado Nunes, Hugo Leite de Farias Brito, Marcia Maria Macedo Lima, Rosana Cipolotti.

**Writing – original draft:** Marcela Sampaio Lima.

**Writing – review & editing:** Érika de Abreu Costa Brito, Hianga Fayssa Fernandes Siqueira, Marceli de Oliveira Santos, Angela Maria da Silva, Marco Antonio Prado Nunes, Hugo Leite de Farias Brito, Marcia Maria Macedo Lima, Rosana Cipolotti, Carlos Anselmo Lima.

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
