## [Decision Letter · Decision Letter 0]

1 Oct 2019

PONE-D-19-19030

Trends in cervical cancer and its precursor forms to evaluate screening policies in a mid-sized Northeastern Brazilian city

PLOS ONE

Dear Dr. Lima,

Thank you for submitting your manuscript to PLOS ONE. After careful consideration, we feel that it has merit but does not fully meet PLOS ONE’s publication criteria as it currently stands. Therefore, we invite you to submit a revised version of the manuscript that addresses the points raised during the review process.

The reviewers raised several concerns that must be addressed for the paper to be publishable. I would like to stress that it is a major revision and I have some doubts that you will be able to address those concerns. My main concern is the first point of the second reviewer. It is also important to provide details of the cancer registry used in the study and describe where the data can be found. Finally, I am also troubled by the lack of specifics on the methods used in the study and its appropriateness for the study. 

We would appreciate receiving your revised manuscript by Nov 15 2019 11:59PM. To enhance the reproducibility of your results, we recommend that if applicable you deposit your laboratory protocols in protocols.io, where a protocol can be assigned its own identifier (DOI) such that it can be cited independently in the future. For instructions see: http://journals.plos.org/plosone/s/submission-guidelines#loc-laboratory-protocols

We look forward to receiving your revised manuscript.

Kind regards,

Gabriel A. Picone

Academic Editor

PLOS ONE

**Journal Requirements:**

**Comments to the Author**

1. Is the manuscript technically sound, and do the data support the conclusions?

Reviewer #1: Partly

Reviewer #2: No

2. Has the statistical analysis been performed appropriately and rigorously? 

Reviewer #1: Yes

Reviewer #2: No

3. Have the authors made all data underlying the findings in their manuscript fully available?

Reviewer #1: No

Reviewer #2: No

4. Is the manuscript presented in an intelligible fashion and written in standard English?

Reviewer #1: No

Reviewer #2: Yes

5. Review Comments to the Author

Reviewer #1: This is a potentially interesting manuscript, if limited in its generalizability due to the specifics of its chosen study population. However, it is in dire need of editing for content and comprehensibility by someone skilled in writing in the English language. As it stands now, the paper is a chore to read and difficult to understand. The comments below are based on my best guess as to what the authors are trying to convey.

Specific comments:

Introduction; last paragraph. Please be clear on what the objective of the study is in as laconic a manner as possible. Do the authors hypothesize that improved (changes) screening rates (in years X,Y,Z) lead to the cancer being identified at earlier stages? That HPV vaccination will lead to reduced incidence rates? Something else?

Lines 70-79. Please be more clear and straightforward: preinvasive lesions are defined as X; invasive as Y; Z is excluded.

Line 81. How did you revise these? Why does it matter if you then exclude?

Lines 82-90. Be very specific in explaining how you calculated age-adjusted rates. What was the standardized population on which you performed the adjustment? You may use a citation for the formula, but not the important details of something central to the paper.

Lines 90-101. Provide more detail on the methods chosen for the study. Less detail on administrative proceedings.

General Comments.

The HDI is mentioned in numerous places in the text, yet its presence is primarily a red herring as it is never used in the analysis. Please avoid providing information that does not contribute to the study. Similarly, I am sure Aracaju is a wunderbar place, but what does its longitude and latitude have to do with cancer?

Provide details on the cancer registry, specifically were any changes made to the way the data was collected (including levels of detail relevant to identifying the study conditions) over the time period included in the study?

Lack of detail on how this cancer registry works brings up the following questions: in situ will progress to invasive cancer at a certain rate. How is this handled by the authors and/or recorded by the registry? in situ patients can also die for numerous reasons, why was mortality among this strata not considered? In general more detail on sample selection is needed... We started with X people, excluded Y people for reasons, final sample sizes were z1 for this an z2 for that category etc.

Is mortality cause-specific (e.g. cause of death documented) or incidence-based (e.g. all mortality after onset).

Table 2 is hard to read; suggest authors move CI below the APC.

An total/combined trend is useful for comparison.

Figures need to be in higher resolution with captions provided for each.

Reviewer #2: Dear authors,

Thanks for the opportunity to review your study. My comments intend to improve your work.

Major comments

1. The research question is to estimate the impact of screening on incidence and mortality from invasive cervical cancer lesions, but there´s no data on screening. I wonder if it is correct to properly determine a causal association in this analysis. The study design seems not appropriate to answer the research question. The conclusion is overestimated.

2. Authors used the ICD D06 for precursor lesion. Is well known that for precursor lesions the ICD N87 should also be evaluated. The figures may be over or underestimated. I wonder that it is not possible to draw a conclusion based on this.

3. It is necessary to clarify more how the ICD C55 was “revised and excluded”. As you are using an extended period for the analysis, it might have influenced the figures. I suggest reading the paper “Disparities in time trends of cervical cancer mortality rates in Brazil. Vale DB et al., Cancer Causes Control. 2016”.

4. Please specify more clearly which world population was used for adjustment (and also include in the tables this information): 1960? 2000? 2010? The reference 7 is a citation, please use an original one. Please clarify why did you choose to use the world population instead of the Brazilian one.

5. The main problem in the trend test used is that the figures are too small, which is expected for a rare condition as cancer… There is a lot of variation among years of the absolute numbers. For example, the number of D06 varies significantly among years… The statistical analysis seems a very superficial way to look at the real figures.

6. There is a conceptual error in the discussion. It is expected that after some years of implementation of a screening program, the precursor lesions rates would decrease, as the prevalent cases will be excluded by treatment. Authors should be more careful with the interpretation of this results.

7. Did you notice that the figures for incidence were very different in 2008 and 2009? What might have happened? Because this difference is crucial to determine the JointPoint analysis presented and, as the absolute figures are too small, I wonder if it is appropriate to use a Joint analysis in this case. I think only the entire period tendency analysis would be adequate.

8. Figures are too small for analysis by age-groups. It is not consistent.

9. In the discussion, authors should include some data on screening in the region during the period.

10. Authors use just a few national/regional references to discuss their results. I think it is important to use more local data to validate the study.

11. Discussion is long and unfocused.

13. Reference 6 – please use the more updated guideline (2016).

6. PLOS authors have the option to publish the peer review history of their article (what does this mean?). If published, this will include your full peer review and any attached files.

Reviewer #1: No

Reviewer #2: Yes: Diama Bhadra Vale

---

## [Author Response · Author response to Decision Letter 0]

27 Mar 2020

Reviewer #1

I have added a new year of data and recalculated trends

- I have submitted to English language edition

- Introduction. I tried to respond to the reviewer’s queries

- Line 70-79. Corrected

- Line 81. The head of the cancer registry and senior author (a surgical oncologist) CAL went back to records and pathology reports to resolve any conflicting registration. That was done to certify all cases were included.

- Line 82-90. Ok, more details were added on how age-specific rates were calculated

- Line 90-101. Ok, I hope I have provided the required information

General comments:

- About HDI and coordinates. Accepted

- Details on the cancer registry. Data collection was modified to include laboratory databases more recently, instead of calling on laboratories to gather data; but not the consistency of case catchment. The cancer registry data has been submitted and approved for Cancer Incidence in 5 continents X and XI, and Concord-2 and -3.

- About in situ. Cases are registered as in situ if no other diagnoses were identified as invasive in less than one year; otherwise, both in situ and invasive were registered. For the present paper, I decided not to consider assess mortality for carcinoma in situ because the impact on mortality would be minimal and, if present, would be mostly due to other causes.

- We assessed al cause mortality from the official Mortality Information System

- Table 2 was modified

- Figures in high resolution were provided

Reviewer #2

1. Actually I tried to estimate the impact of screening on incidence and subsequently on mortality; however, data on screening, other that of SISCOLO, is not available, which is included in the registry database

2. ICD N87 is not reported. Some pathologists use the term high grade/severe dysplasia interchangeably with carcinoma in situ. High grade dysplasia is only reported as carcinoma in situ when the pathologist considers them to be the same – it is documented as carcinoma in situ

3. ICD C55 was revised to make sure topography and histology was not related to the uterine cervix. In case, they were from other parts of the uterus, they were excluded from analysis

4. More detail on calculation of age standardized data was included – the standardization by the direct method was used (Segi, 1960) which defines the number of individuals by 5-year age groups to a total of 100,000. We preferred not to use the Brazilian population because if so comparisons would be only in Brazil and the world population is thoroughly used. 

5. I agree that figures are small but trend analysis using the Joinpoint Regression Program is validated worldwide

6. I tried to overcome that

7. Yes, probably due to under catchment of cases, and that possibly made the following year present a higher number of cases

8. Ok, that’s the way it is

9. Unluckily, we have on screening was collected by SISCOLO, which is one of our data sources

10. I have tried but local data is scarce

11. I have reorganized discussion

12. Ok

- I have added an additional year of data and recalculated trends

---

## [Decision Letter · Decision Letter 1]

10 Apr 2020

PONE-D-19-19030R1

Trends in cervical cancer and its precursor forms to evaluate screening policies in a mid-sized Northeastern Brazilian city

PLOS ONE

Dear Dr. Lima,

Thank you for submitting your manuscript to PLOS ONE. After careful consideration, we feel that it has merit but does not fully meet PLOS ONE’s publication criteria as it currently stands. Therefore, we invite you to submit a revised version of the manuscript that addresses the points raised during the review process.

I agree with the reviewers that the paper is much improved. One of the reviewers is ready to accept the paper, while the other has minor comments. These comments are easy to address, and I hope that you can respond to them right away so I can accept the paper.

We would appreciate receiving your revised manuscript by May 25 2020 11:59PM. To enhance the reproducibility of your results, we recommend that if applicable you deposit your laboratory protocols in protocols.io, where a protocol can be assigned its own identifier (DOI) such that it can be cited independently in the future. For instructions see: http://journals.plos.org/plosone/s/submission-guidelines#loc-laboratory-protocols

We look forward to receiving your revised manuscript.

Kind regards,

Gabriel A. Picone

Academic Editor

PLOS ONE

Reviewers' comments:

Reviewer's Responses to Questions

**Comments to the Author**

1. If the authors have adequately addressed your comments raised in a previous round of review and you feel that this manuscript is now acceptable for publication, you may indicate that here to bypass the “Comments to the Author” section, enter your conflict of interest statement in the “Confidential to Editor” section, and submit your "Accept" recommendation.

Reviewer #1: (No Response)

Reviewer #2: All comments have been addressed

2. Is the manuscript technically sound, and do the data support the conclusions?

Reviewer #1: Yes

Reviewer #2: Yes

3. Has the statistical analysis been performed appropriately and rigorously? 

Reviewer #1: Yes

Reviewer #2: Yes

4. Have the authors made all data underlying the findings in their manuscript fully available?

Reviewer #1: Yes

Reviewer #2: Yes

5. Is the manuscript presented in an intelligible fashion and written in standard English?

Reviewer #1: No

Reviewer #2: Yes

6. Review Comments to the Author

Reviewer #1: The manuscript is much improved.

1. Do not use blue/light blue in figure 3 - hard to tell the difference. Choose a different color.

2. What is meant by "Brazilian health policies need to be re-evaluated"? Be explicit. As is, this advice is very vague.

3. Report the smallest number of cases across the age groups used. Age-adjusting using the world's population gives questionable results when the case numbers in an age group are exceptionally small (should not be an issue, but best to verify, a response in the rebuttal letter is enough - no need to put in text).

4. What is the probability of in situ progressing to invasive tumor? If small the conclusion would be weakened.

5. A second run of English editing is needed. I has improved greatly from the first draft, but still.

Reviewer #2: Dear authors,

You have addressed the issues and improved the manuscript substantially.

Congratulations.

7. PLOS authors have the option to publish the peer review history of their article (what does this mean?). If published, this will include your full peer review and any attached files.

Reviewer #1: No

Reviewer #2: Yes: Diama Bhadra Vale

---

## [Author Response · Author response to Decision Letter 1]

21 Apr 2020

Reviewer #1

1. I believe that remark was for figure 2.

Joinpoint Regression Program won’t allow to change line colors. My preference was to keep the graphic generated by the program. To overcome the difficulty mentioned, I increased the line width. I hope that will do.

2. I included some words, trying to be explicit on that. That can be seen in the marked-up copy.

3. For the sake of synthesis, the authors decided not to report number of cases across age groups, which can be found in the supplementary material file. We believed that was nor important, but actually, we calculated age-specific rates instead.

4. I included some words trying to respond to that remark. That can be seen in the marked-up copy.

5. We submitted the manuscript to Edanz Group for a second run of English editing. 

Reviewer #2:

No additional revision was required.

---

## [Editor Report · Decision Letter 2]

5 May 2020

Trends in cervical cancer and its precursor forms to evaluate screening policies in a mid-sized Northeastern Brazilian city

PONE-D-19-19030R2

Dear Dr. Lima,

We are pleased to inform you that your manuscript has been judged scientifically suitable for publication and will be formally accepted for publication once it complies with all outstanding technical requirements.

With kind regards,

Gabriel A. Picone

Academic Editor

PLOS ONE
---

## [Editor Report · Acceptance letter]

8 May 2020

PONE-D-19-19030R2 

Trends in cervical cancer and its precursor forms to evaluate screening policies in a mid-sized Northeastern Brazilian city 

Dear Dr. Lima:

I am pleased to inform you that your manuscript has been deemed suitable for publication in PLOS ONE. Congratulations! Your manuscript is now with our production department. 

With kind regards,

on behalf of

Dr. Gabriel A. Picone 

Academic Editor

PLOS ONE